# CSF1R Ligands Expressed by Murine Gliomas Promote M-MDSCs to Suppress CD8^+^ T Cells in a NOS-Dependent Manner

**DOI:** 10.3390/cancers16173055

**Published:** 2024-09-01

**Authors:** Gregory P. Takacs, Julia S. Garcia, Caitlyn A. Hodges, Christian J. Kreiger, Alexandra Sherman, Jeffrey K. Harrison

**Affiliations:** Department of Pharmacology & Therapeutics, College of Medicine, University of Florida, Gainesville, FL 32610, USAjuliagarcia@ufl.edu (J.S.G.); caitlyn.hodges@ufl.edu (C.A.H.);

**Keywords:** glioma, immune, myeloid, MDSC, T cell, suppression, CSF, NOS, chemokine receptor

## Abstract

**Simple Summary:**

Currently, there are no effective therapies for glioblastoma. Infiltrating myeloid cells contributes significantly to the immune suppressive tumor microenvironment that is characteristic of GBM. Monocytic myeloid-derived suppressor cells are chief immune suppressive cells found in the glioma microenvironment. Understanding the mechanisms of M-MDSC differentiation and T-cell suppression is imperative for generating therapies that target this tumor-supportive cell population. In this study, we found that glioma-secreted CSF1R ligands, M-CSF and IL-34, promote M-MDSCs to suppress CD8 T cells. These M-MDSCs partially utilize nitric oxide synthase to illicit their suppressive activity. However, spatial RNAseq points to glioma microenvironment niches driving M-MDSC heterogeneity. Our findings identify key regulators of differentiation and suppressive mechanisms of M-MDSCs and confirm the importance of targeting this cell population in glioma.

**Abstract:**

Glioblastoma (GBM) is the most common malignant primary brain tumor, resulting in poor survival despite aggressive therapies. GBM is characterized by a highly heterogeneous and immunosuppressive tumor microenvironment (TME) made up predominantly of infiltrating peripheral immune cells. One significant immune cell type that contributes to glioma immune evasion is a population of immunosuppressive cells, termed myeloid-derived suppressor cells (MDSCs). Previous studies suggest that a subset of myeloid cells, expressing monocytic (M)-MDSC markers and dual expression of chemokine receptors CCR2 and CX3CR1, utilize CCR2 to infiltrate the TME. This study evaluated the mechanism of CCR2^+^/CX3CR1^+^ M-MDSC differentiation and T cell suppressive function in murine glioma models. We determined that bone marrow-derived CCR2^+^/CX3CR1^+^ cells adopt an immune suppressive cell phenotype when cultured with glioma-derived factors. Glioma-secreted CSF1R ligands M-CSF and IL-34 were identified as key drivers of M-MDSC differentiation while adenosine and iNOS pathways were implicated in the M-MDSC suppression of T cells. Mining a human GBM spatial RNAseq database revealed a variety of different pathways that M-MDSCs utilize to exert their suppressive function that is driven by complex niches within the microenvironment. These data provide a more comprehensive understanding of the mechanism of M-MDSCs in glioblastoma.

## 1. Introduction

Glioblastoma (GBM) is a highly invasive, aggressive, and common WHO Grade IV brain tumor, representing 50.9% of all malignant CNS tumors [1]. The current standard of care therapy for GBM consists of maximal surgical resection followed by radiotherapy plus concomitant temozolomide (TMZ) chemotherapy [2,3]. Despite this intensive treatment regimen, prognosis for GBM patients remains poor, with a five-year survival rate of 6.9% and a median survival length of only 8 months [1]. The immunosuppressive tumor microenvironment (TME) of GBM is not only a fundamental driver of the poor outcomes seen in treatment, but also a target of interest in the context of immunotherapy [4,5,6]. Immune checkpoint inhibitors, especially those that target the PD-L1/PD-1 axis, have shown efficacy in other cancers, but GBM remains an exception [7,8,9]. A key contributor to the immunosuppressive TME seen in GBM is a population of infiltrating myeloid cells called myeloid-derived suppressor cells (MDSCs) [10,11,12].

MDSCs are a heterogeneous population of bone marrow-derived cells consisting of monocytic (M)-MDSCs and polymorphonuclear (PMN)-MDSCs subtypes [13]. Murine MDSCs are characterized by their expression of respective lineage markers: CD45^+^CD11b^+^Ly6C^hi^Ly6G^−^ (M-MDSCs) and CD45^+^CD11b^+^Ly6C^lo^Ly6G^+^ (PMN-MDSCs) [14,15]. MDSCs in humans are identified by expression of general lineage markers CD33^+^ MHC-II^−^, with specific subtypes being defined by expression of CD11b^+^CD14^+^CD15^−^ (M-MDSCs) and CD11b^+^CD14^+^CD66b^+^ (PMN-MDSCs) [14]. M-MDSCs are especially of interest in GBM as they are highly prevalent while PMN-MDSCs are minimally present in the TME [13,15,16]. Our previous work highlighted the presence of a population of myeloid cells in the glioma TME that expresses the chemokine receptors CCR2^+^ and CX3CR1^+^. These bone marrow-derived CCR2^+^/CX3CR1^+^ cells also co-express markers that correspond to M-MDSCs, namely, CD45^+^, CD11b^+^, Ly6C^hi^, and Ly6G^−^, migrate to glioma-secreted CCL2 and CCL7, and suppress the proliferation and IFN-γ secretion of both CD4^+^ and CD8^+^ T cells [17,18].

While our previous research highlighted both the migratory and immunosuppressive nature of these CCR2^+^/CX3CR1^+^ M-MDSCs, it remained unknown what factors induce their immunosuppressive phenotype and the mechanisms that M-MDSCs utilize to suppress T cells in GBM. We hypothesized that soluble factors secreted by the tumor promote the differentiation and expansion of the M-MDSC population from naïve bone marrow. In other diverse contexts, various markers of inflammation induce and enhance the immunosuppressive phenotype of MDSCs, including IL-1β, IFN-γ, IL-6, GM-CSF, G-CSF, and M-CSF [19,20,21]. Monocyte-colony stimulating factor (M-CSF/CSF1) and IL-34 both bind to colony-stimulating factor-1 receptor (CSF-1R), a receptor tyrosine kinase [22]. Inhibition of CSF-1R has been targeted in several cancers, including melanoma, lung carcinoma, and prostate cancer, to neutralize M-MDSCs [23,24,25]. Of the cytokines upregulated in the glioma TME, we identified Colony Stimulating Factors (CSFs) to be sufficient for M-MDSC differentiation. Compared to other CSF ligands, CSF1 is upregulated in both murine and human GBM. We also report that IL-34 is expressed in both murine and human GBM. This highlights the instrumental role of glioma-derived CSF1 and IL-34 in the differentiation of M-MDSCs, as blocking the CSF1R axis prevents expansion of the M-MDSC population in whole bone marrow.

A wide variety of mechanisms of MDSC immunosuppression of T cells have also been explored, including depletion of amino acids such as arginine and tryptophan, production of reactive oxygen species, adenosine production, and expression of PD-L1 and CTLA-4 [26,27]. Herein, we investigated the diversity in potential immunosuppressive pathways utilized by MDSCs by mining the Ivy glioblastoma Atlas Project database. We highlight two pathways that were upregulated in different regions of the tumor: iNOS and adenosine. Inducible nitric oxide synthase (iNOS) is a key synthesizer of nitric oxide, a free radical molecule with diverse impacts on T cells, including induction of apoptosis, upregulation of death receptors, and DNA damage [28,29]. Adenosine is generated from ATP through a multistep process involving PGE2, CD39, and CD73: CD39 converts ATP to AMP, which is then converted to adenosine by CD73 [30,31]. PGE2 also plays a key role in inducing CD73 expression in M-MDSCs [31]. Specifically, we identify that iNOS is expressed on some M-MDSCs and that iNOS inhibition recovers the proliferation of CD8^+^ T cells in the presence of M-MDSCs. Additionally, a subset of M-MDSCs in murine GL261 and KR158B gliomas expresses CD39 and, to a lesser extent, CD73, both of which are involved in the production of adenosine. PGE2 is also expressed by both of these murine glioma cell lines. Overall, these data highlight the intricate and interdependent relationship between M-MDSCs and GBM, furthering understanding of the complex mechanisms involved in the licensing and immunosuppressive phenotype of M-MDSCs within the context of glioblastoma.

## 2. Materials and Methods

### 2.1. Animals

Wildtype C57BL/6 mice and Ccr2^RFP/WT^/Cx3cr1^GFP/WT^ mice were utilized. Ccr2^RFP/WT^/Cx3cr1^GFP/WT^ were generated by breeding Ccr2-deficient (Ccr2RFP/RFP[B6.129(Cg)-Ccr2tm2.1Ifc/J]) mice with Cx3cr1-deficient (Cx3cr1GFP/GFP[B6.129P-Cx3cr1tm1Litt/J]) mice. Wildtype C57BL/6, Ccr2-deficient, and Cx3cr1-deficient mice were purchased from The Jackson Laboratory. All surgical protocols and animal housing procedures followed the University of Florida Institutional Animal Care and Use Committee guidelines.

### 2.2. Orthotopic Implantation

Prior to cell injection, animals were administered an isoflurane anesthetic and an analgesic. Once under anesthesia, with the surgical site prepared, a 2-to-3-mm incision was made at the midline of the skull. A stereotaxic apparatus (Stoelting) secured the mice, and a Hamilton syringe was positioned 2 mm lateral from the bregma. KR158B or GL261 glioma cells (7.5 × 10^4^ in a total volume of 2 μL) were injected 3 mm deep into the right cerebral hemisphere using an automated microfluidic injection system (Stoelting) at a rate of 1 μL/min; cells were suspended in a 1:1 ratio of methylcellulose to PBS. Once injected, the needle was slowly retracted, with the surgical site closed via suture and bone wax before animals were placed in a warm cage for postsurgical monitoring.

### 2.3. INOS Immunohistochemistry

After euthanasia via isoflurane overdose, a transcardial perfusion of 20 mL of 4.0% paraformaldehyde (PFA) was performed, using a 10 mL syringe and 25 G winged infusion set. Mouse brains were removed, soaked in 4.0% PFA for 1 h, and transferred to a 30% sucrose solution for 24 h. Brains were snap-frozen using liquid-nitrogen-chilled 2-methylbutane. Brains were embedded in optimal cutting temperature compound (OCT) and sectioned in 10 μm thick sections on a Leica Biosystems Cryostat. Sections were mounted onto microscope slides and dried overnight at 4 °C. Sections were brought to room temperature, washed 3 times in 1× PBS for 5 min, and permeabilized in 0.1% Triton X-100 in PBS for 5 min. After washing 3 times in 1× PBS for 5 min, sections were blocked in 10% NGS in PBS for 1 h, washed 3 more times in 1× PBS, blocked in 10% BSA in PBS for 1 h, and washed 3 more times in 1× PBS for 5 min. Permeabilization and both blocking steps took place in a humidified chamber at room temperature. Sections were stained with rabbit polyclonal anti-iNOS antibody (Abcam, Waltham, MA, USA, ab15323) at a 1/200 dilution with 5% NGS in PBS overnight in a humidified chamber at 4 °C. Samples were washed 3 times in 1× PBS for 5 min, stained with goat anti-rabbit secondary antibody (Invitrogen, Waltham, MA, USA, a21244) for 1 h in a humidified chamber at room temperature, and washed 3 more times in 1× PBS. Samples were counterstained with antifade mounting medium with DAPI (Vectashield, Vector Laboratories, Newark, CA, USA), imaged with an inverted Nikon TiE-PFS-A1R confocal microscope, and post-processed with Nikon Elements software v5.21.

### 2.4. Tissue Processing

Blood, bone marrow, spleen, and tumor were isolated and processed as described in previous work [18]. Mice were euthanized via isoflurane overdose. The right atrium was snipped, allowing blood to pool in the chest cavity. A total of 200 μL of blood was collected by using a 1 mL syringe coated in EDTA. Blood was transferred into a 1.5 mL tube filled with 100 μL of EDTA and centrifuged (380× *g*, 5 min, RT); plasma was discarded. A transcardial perfusion of 20 mL of 0.9% saline was then performed, using a 10 mL syringe and 25 G winged infusion set. Femurs, tibiae, humeri, and the pelvis were harvested and cleaned of fat and muscle. One end of each bone was snipped, and bones were placed into 0.5 mL centrifuge tubes that had been pierced on the bottom by an 18 G needle. The 0.5 mL tubes with the bones (2 bones per tube) were nested inside 1.5 mL tubes filled with 100 μL PBS and centrifuged (5700× *g*, 20 s, RT) to extract the bone marrow cells. Spleens were isolated, trimmed of fat, placed in a petri dish with 1 mL PBS, and minced with a razor blade. Minced spleens in PBS were transferred to a 15 mL conical tube with an additional 4 mL of PBS and mechanically dissociated via passage through an 18 G needle and 5 mL syringe 20 times. Splenocytes were then centrifuged (380× *g*, 5 min, RT). Blood, bone marrow cells, and splenocytes were resuspended in 1 mL Ammonium–Chloride–Potassium (ACK) Lysis buffer (Gibco, Invitrogen) for 5 min on ice to lyse red blood cells. Lysis was quenched with 5 mL fluorescence-activated cell-sorter (FACS) washing buffer (1% FBS and 2 mM EDTA in PBS). Blood underwent a total of 3 cycles of ACK lysis and FACS buffer quenching, with centrifugation (380× *g*, 5 min, RT) between each cycle. Single-cell suspensions of blood cells, bone marrow cells, and splenocytes were then strained through a 40 μm cell strainer. Samples were centrifuged (380× *g*, 5 min, RT), resuspended in 1 mL FACS buffer, and transferred to 1.5 mL microcentrifuge tubes. Brains were removed and tumors were mechanically excised and minced with a razor blade. Minced tumors were transferred to a 50 mL conical tube with ice-cold Accumax dissociation solution (Innovative Cell Technologies, San Diego, CA, USA) and incubated for 5 min at 37 °C. Cells were returned to room temperature for 5 min and further dissociated through passage through a P1000 pipette with the end of the tip cut off. The suspension was passed through a 70 μm cell strainer with a pestle, washed with 9 mL of ice-cold FACS buffer, and centrifuged (380× *g*, 5 min, 4 °C). Cells were resuspended in 4 mL of 70% Percoll solution (70% Percoll and 10% 10× PBS in RPMI-1640 cell medium), which was layered with an 18 G needle beneath 4 mL of 37% Percoll solution (37% Percoll and 10% 10× PBS in RPMI-1640 cell medium). Samples were centrifuged (500× *g*, 30 min, RT) and cells at the interface between Percoll layers were collected and transferred into a 1.5 mL microcentrifuge tube.

### 2.5. Luminex Cytokine Analysis

Eve Technologies (Calgary, AB, Canada) Mouse Cytokine/Chemokine 44-Plex Discovery Assay^®^ Array (MD44) services were utilized to evaluate cytokine levels in naïve brain and KR158B tumors. KR158B tumors were implanted in wildtype C57BL/6 mice. Two weeks post-implantation, a brain tumor and an identical region of naïve brain (from a different mouse) were dissected. The tissue was lysed using 1 mL NP40 cell lysis buffer and 1× Halt Protease inhibitor cocktail. The tissue was homogenized using a glass pestle and left on ice for 20 min. Samples were centrifuged (10,000× *g*, 10 min, 4 °C). A sample (75 μL) was aliquoted and sent for Luminex while 10 μL was used to measure total protein concentration (Bradford Assay). Samples were snap-frozen in liquid nitrogen and stored at −80 °C until used. The concentration of each cytokine per sample was normalized to total protein. Data are presented as Log2 fold change compared to an average of naïve brain samples.

### 2.6. MDSC Induction

For MDSC induction by KR158B-conditioned media, bone marrow cells from Ccr2RFP/WT/Cx3cr1GFP/WT mice were plated at a density of 400,000 cells/cm^2^ at a concentration of 1000 cells/μL in complete RPMI (RPMI + 10% FBS + 2 mM L-Glutamine) and increasing concentrations of KR158B-conditioned media (25%, 50%, and 75%). KR158B-conditioned media was collected from 3 million KR158B glioma cells that were plated in 25 mL of complete DMEM media in T182cm2 flask and grown for 3 days. For MDSC induction by cytokines, 750,000 bone marrow cells from wildtype C57BL/6 mice were plated in 750 μL of complete RPMI in a 24-well plate. Cytokines GM-CSF, M-CSF, G-CSF, TGF-β, IFNγ, IL-1β, IL-2, IL-6, IL-12p40, IL-13, and LIF were added to individual wells at a concentration of 40 ng/mL. Wells with 75% KR-conditioned media served as a positive control for MDSC induction. For both experiments, cells were incubated for 3 days in a humidified incubator at 37 °C with 5% CO_2_. Suspended cells were then collected and adherent cells were scraped with a cell scraper (Thermo Scientific, Waltham, MA, USA) and washed in PBS. Flow cytometry was then performed for both experiments and immunocytochemistry was performed for the KR-conditioned media experiment. Cytokines used for induction can be found in the Appendix A.

### 2.7. Bone Marrow Culture

Bone marrow-derived cells from wildtype C57BL/6 mice were plated at a density of 400,000 cells/cm^2^ and a concentration of 1000 cells/μL in 50% complete RPMI and 50% KR158B-conditioned media, supplemented with 40 ng/mL GM-CSF (R&D 415-ML) and 40 ng/mL IL-6 (R&D 406-ML). Cells were grown for 3 days in a humidified incubator at 37 °C with 5% CO_2_. On day 3, suspended cells were collected, remaining cells were washed in PBS and scraped using a cell scraper (Fisher), and all contents were combined in a 50 mL conical tube. Cells were collected via centrifugation (380× *g*, 5 min, 4 °C) and either subjected to flow cytometry (see “Section 2.9”) or utilized in a T-cell suppression assay (see “Section 2.11”).

### 2.8. Glioma Cell Culture

KR158B glioma cells were cultured in Dulbecco’s Modified Eagle Medium (DMEM), supplemented with 1% penicillin–streptomycin and 10% fetal bovine serum (FBS). GL261 glioma cells were cultured in RPMI-1640 medium, supplemented with 1% penicillin–streptomycin, 10% FBS, and 4 mM L-Glutamine. All cells were grown in a humidified incubator at 37 °C with 5% CO_2_. DMEM and penicillin–streptomycin were purchased from Invitrogen; FBS was purchased from Thermo Scientific (Waltham, MA, USA).

### 2.9. Flow Cytometry

Single-cell suspensions were prepared from tissues as described above (see “Section 2.4”), centrifuged (380× *g*, 5 min, RT), and resuspended in FCR block for 5 min. Cells were stained for markers of interest (Appendix A) for 30 min at 4 °C. Staining was quenched with FACS buffer. Samples were then centrifuged (380× *g*, 5 min, RT) and stained with a viability dye (15 min, RT), which was again quenched with FACS buffer. For intracellular staining, an Intracellular Fixation & Permeabilization Buffer Set was used (eBioscience, Waltham, MA, USA, catalog # 88-8824-00). Viability dye was quenched with 500 μL of PBS. Samples were then incubated with 100 μL of fixation buffer (30 min, RT). The fixation buffer was quenched with 1 mL of permeabilization buffer and centrifuged twice (500× *g*, 5 min, RT). Cell pellets were resuspended in 100 μL of permeabilization buffer and stained with 1 μL of iNOS antibody (30 min, RT). Samples were resuspended twice in 1 mL of permeabilization buffer and centrifuged twice (500× *g*, 5 min, RT). All samples were analyzed by single-color compensation on a Sony SP6800 spectral analyzer. Data were quantified using FlowJo V10.8.1 (BD Biosciences, Franklin Lakes, NJ, USA). Flow gating strategy and antibodies can be found in the Appendix A.

### 2.10. ELISA Assays

For M-CSF (R&D Systems, Minneapolis, MN, USA, MMC00B), GM-CSF (R&D Systems, MGM00), IL-34 ELISA (R&D Systems, M3400) ELISA assays, increasing quantities of KR158B glioma cells (5 × 10^4^, 1 × 10^5^, and 2 × 10^5^) were plated in 96-well plates. For the PGE2 ELISA (R&D Systems, KGE004B), both KR158B and GL261 glioma cells were plated at the same concentrations as above in 96-well plates. All ELISA assays were performed per manufacturers’ specifications and read on a Biotek Synergy 2 plate reader.

### 2.11. T-Cell Suppression Assay

Following a 3-day culture, M-MDSCs expanded from bone marrow were collected (see “Section 2.7”) and sorted via magnetic bead isolation (Miltenyi Biotec) based on the manufacturer’s protocols. Splenocytes were isolated (see “Section 2.4”) and subjected to Pan T-cell magnetic bead isolation (Miltenyi Biotec) based on the manufacturer’s protocols. Isolated CD4^+^/CD8^+^ T cells were collected via centrifugation and resuspended in PBS at a density of 1 × 10^6^ cells/mL. T cells were then incubated with 1 μL of CellTrace FarRed Cell Proliferation dye (ThermoFisher, Waltham, MA, USA, C34564) per 1 × 10^6^ cells (20 min, RT) and quenched with 5 times the sample volume in complete RPMI. T cells are centrifuged (380× *g*, 5 min, 4 °C) and resuspended in complete RPMI at a density of 1 × 10^6^ cells/mL. Stained T cells were re-counted to ensure the correct amount to be activated. Dynabeads Mouse T-Activator CD3/CD28 beads (Thermofisher, 11452D) were washed in complete RPMI and added to stained T cells at a 2:1 (activating bead:T cell) ratio. Unstained/unstimulated and stained/unstimulated controls were retained at each step. In a round-bottom 96-well plate, T cells were plated at a density of 1 × 10^4^ cells/well, with M-MDSCs added at 1:4, 1:2, and 1:1 ratios of MDSCs:T cells. Inhibitors were added at the following concentrations: L-NMMA (500 uM), nor-NOHA (300 uM), 1-Methyl-D-tryptophan (500 uM), α-IFNGR2 (5 ug/mL), and αPD-L1 (5 ug/mL). After incubation at 37 °C for 3 days, wells were scraped and washed twice in PBS. Well contents were transferred to centrifuge tubes and activating beads were removed by the Dynamag-2 (Thermofisher, 12321D). Cells were collected by centrifugation and stained for CD3, CD4, and CD8 (Appendix A) for flow cytometry (see “Section 2.9”). All conditions were run in triplicate, with technical triplicates averaged prior to statistical analysis.

### 2.12. Human GBM Expression: Database

Ivy glioblastoma Atlas Project database was used to investigate spatial regulation of immune suppressive factors in human glioblastoma. The Ivy Glioblastoma Atlas Project is a resource for exploring the anatomic and genetic basis of glioblastoma at the cellular and molecular levels. Briefly, normalized gene-level FPKM values for all samples were downloaded. Interest genes were queried, and Z-scores were calculated for each gene transcript level/region of interest pair. Z scores were grouped according to region of interest (leading edge, infiltrating tumor, cellular tumor, peri necrotic zone, perivascular zone). The OncoDB database was queried for expression of CSFR ligands (CSF1, CSF2, CSF3, and IL-34). The GBM TCGA dataset was used for the analysis and compared to normal tissue from the GTEx database. Single Cell Portal database from Broad Institute was accessed for evaluation of CSF ligand expression. The “Programs, Origins, and Niches of Immunomodulatory Myeloid Cells in Human Gliomas” dataset was used for analysis. Eighty-five human tumors and 515,782 cells were analyzed.

### 2.13. Statistical Analysis

Multiple t-tests, one-way ANOVA, and two-way ANOVA analyses were performed in GraphPad Prism version 9.3.1 to determine statistically significant differences between groups. Multiple comparisons were corrected with the recommended Dunnett multiple comparison test. A *p*-value < 0.05 was considered significant and is indicated by symbols depicted in the figures, figure legends, and text.

## 3. Results

### 3.1. KR158B Glioma-Conditioned Media Enriches CCR2+/CX3CR1+ M-MDSCs

MDSCs arise from a common myeloid progenitor. During tumor progression, naïve monocytes differentiate into Ly6C^hi^ M-MDSCs [32]. Understanding the factors present within the glioma that drive this process is needed to better understand how to target M-MDSCs. Tumor-derived factors (i.e., IL-6, IL-10, IL-1β, PGE2, VEGF, GM-CSF, M-CSF, and IFN-γ) are implicated in the induction of MDSCs [32]. Our previous work established that myeloid cell populations that co-express the chemokine receptors CCR2 and CX3CR1 also express markers that are associated with M-MDSCs (i.e., CD45, CD11b, Ly6C^hi^, Ly6G^neg^). These CCR2^+^/CX3CR1^+^ cells are present in both bone marrow and the TME of KR158B and GL261 tumors. We first hypothesized that glioma cells induce the differentiation process of CCR2^+^/CX3CR1^+^ MDSCs by secreting soluble factors that trigger M-MDSC differentiation. To test this hypothesis, we cultured naïve bone marrow cells in the presence of glioma-conditioned media. Naïve, non-glioma bearing, *Ccr2^WT/RFP^*/*Cx3cr1^WT/GFP^* mouse bone marrow was extracted and plated on glass coverslips in the presence of RPMI (control) or 75% KR158B glioma-conditioned media. After 3 days, coverslips were fixed, stained with DAPI, and imaged. The 75% glioma-conditioned media condition resulted in a higher density of CCR2^RFP/WT^/CX3CR1^GFP/WT^ expressing cells (Figure 1A). Parallel flow cytometry was conducted to quantitate the percentage of CCR2/CX3CR1 expressing cells that also displayed markers of M-MDSCs (Ly6G^−^/Ly6C^hi^). The CCR2^+^/CX3CR1^+^ population increased from 5.1% of live cells in the RPMI control to 20.7%, 34.9%, and 42.2% of live cells in increasing concentrations of KR158B-conditioned media (25%, 50%, and 75% KR158B-conditioned media, respectively) (Figure 1B,D). The percentage of these chemokine receptor-expressing cells that displayed markers of M-MDSCs also increased dose-dependently (31.7%, 56.5%, 57.9%, 75.6%) in the presence of KR158B-conditioned media (Figure 1C,E). An increase in CX3CR1^GFP^ MFI was also observed (Figure 1F). These data establish that KR158B glioma cells directly promote naïve bone marrow cells into CCR2/CX3CR1 M-MDSCs.

### 3.2. Glioma-Secreted CSF1R Ligands Drive M-MDSC Differentiation

To identify the factors in the glioma microenvironment that drive M-MDSC differentiation, we conducted a Luminex-based cytokine analysis to screen for potential mediators of M-MDSC licensing. Wildtype C57BL/6 mice were implanted with KR158B glioma cells or left naïve. Gliomas and an identical region in the naïve brain were extracted and prepared for Luminex analysis. Chemokines associated with T cell trafficking (CCL3, CCL5, CXCL9, and CXCL10) were all upregulated in KR158B glioma tissue compared to naïve tissue. CCL2, previously implicated in M-MDSC trafficking [18], was also upregulated in the TME. CSFR ligands (GM-CSF, G-CSF, and M-CSF) have been reported as key regulators of MDSC differentiation, and all of these CSFR ligands were differentially upregulated in the KR158B gliomas (Figure 2A).

To further investigate the KR158B-secreted factors that drive M-MDSC differentiation, all of the differentially upregulated cytokines were tested for their ability to expand M-MDSCs from whole bone marrow. Briefly, whole bone marrow was extracted and plated in KR158B glioma-conditioned media or exogenous cytokines. After 3 days, cells were collected and stained for MDSC markers Ly6C and Ly6G (Figure 2B). As expected, KR158B-conditioned media, GM-CSF, G-CSF, and M-CSF expanded M-MDSCs (percentage and numbers) from whole bone marrow (Figure 2B,C). GM-CSF yielded the highest number of MDSCs (monocytic and polymorphonuclear) compared to the other cytokines evaluated. However, the M-CSF condition appeared most similar to the effect of the KR158B-conditioned media. PMN-MDSCs, macrophages, and dendritic cells were also profiled. PMN-MDSCs were only expanded under GM-CSF and G-CSF conditions. Interestingly, KR-conditioned media resulted in lower numbers of PMN-MDSCs compared to RPMI control Appendix A. GM-CSF, M-CSF, and KR-conditioned media expanded the percentage and total number of F4/80 macrophages (Appendix A) while GM-CSF increased the number of CD11c positive dendritic cells (Appendix A).

### 3.3. CSF Ligands Are Regulated by Glioma Cells in the Glioma Microenvironment

Since exogenous M-CSF impacted M-MDSC differentiation similar to KR158B-conditioned media, protein expression of CSF ligands, namely, M-CSF (CSF1R ligand), GM-CSF (CSF2R ligand), and IL-34 (the other CSF1R ligand), was evaluated in KR158B-conditioned media. KR158B glioma cells were plated at three densities (50k, 100k, and 200k) and conditioned media was subsequently analyzed after 24 h for M-CSF, IL34, and GM-CSF. M-CSF was secreted at high levels (mean 1333 pg/mL 200k cells) compared to GM-CSF and IL-34. Interestingly, IL-34 was detected at higher cell densities (mean 8.2 pg/mL 200k cells) while GM-CSF was barely detectable (mean 0.8 pg/mL 200k cells) (Figure 3A). To expand on these findings, human GBM expression of CSF ligands was extracted from the OncoDB database. TCGA dataset was queried for GBM gene expression and results were compared to normal brain tissue from GTEx. CSF-1 (M-CSF) was differentially upregulated in the GBM microenvironment compared to the normal brain. CSF-1 was found at the highest level compared to the other CSF ligands. IL-34 was present but at lower levels compared to normal tissue. CSF2 (GM-CSF) and CSF3 (G-CSF) were expressed at very low levels in normal and GBM tissue (Figure 3B). In an attempt to distinguish which cells within the human glioma microenvironment are regulating these factors, the Single Cell Portal database from Broad Institute was accessed for evaluation of CSF ligand expression. The “Programs, Origins, and Niches of Immunomodulatory Myeloid Cells in Human Gliomas” dataset was used for analysis. Multiple human tumors (85) and cells (515,782) were analyzed. CSF-1 (M-CSF) and IL34 were determined to be expressed by the malignant cell population and other tumor-associated cells. CSF-2 and CSF-3 (GM-CSF and G-CSF) were expressed at low levels and did not map to malignant cell types (Figure 3C). Taken together, these data establish that M-CSF and IL-34 are secreted by glioma cells in both human and murine GBM.

### 3.4. Inhibition of the CSF1R Axis Blocks Glioma-Mediated M-MDSC Differentiation

To implicate M-CSF and its cognate receptor CSF1R, neutralizing antibodies and small molecule antagonists were used to block KR158B-mediated M-MDSC differentiation. Inhibiting M-CSF alone (1 ug/mL of anti-M-CSF Ab) did not result in a complete block of M-MDSC differentiation (Figure 4A, Appendix A). This suggested that IL-34, the other CSF1R ligand, may compensate in the absence of functional M-CSF. We utilized a combination approach to block both IL-34 and M-CSF simultaneously in KR158B-conditioned media. Like M-CSF inhibition, IL-34 neutralization alone (1 ug/mL of anti-IL-34) was unable to completely prevent M-MDSC differentiation from glioma-conditioned media (Figure 4A, Appendix A). However, neutralization of both M-CSF and IL-34 in combination (1 ug/mL anti-M-CSF and 0.5 or 1 μg/mL anti-IL-34) yielded a reduction greater than either of the mono treatments (Figure 4A, Appendix A). To validate that the CSF1 receptor was involved in M-MDSC differentiation, the CSF1R antagonist Pexidartinib was utilized. CSF1R inhibition resulted in a dose-dependent decrease in M-MDSC differentiation that led to a complete block at 320 nM (Figure 4B, Appendix A). PMN-MDSCs were also profiled. Neutralizing M-CSF and IL-34 or blocking CSF1R did not result in changes in PMN-MDSCs numbers (Figure 2A,B, Appendix A). These data directly implicate CSF1R ligands M-CSF and IL-34 as the main drivers of KR158B-mediated M-MDSC differentiation with no impact on PMN-MDSCs.

### 3.5. CD8+ T Cell Suppressive State Can Be Partially Ameliorated by iNOS Inhibition

M-MDSCs preferentially use nitric oxide, immunosuppressive cytokines (i.e., IL-10 and TGFβ), and inhibitory immune checkpoint molecules, such as PD-L1, to suppress immune responses [32]. Our previous work established that CCR2^+^/CX3CR1^+^ cells display markers of known immune suppressive cell types (M-MDSCs) [17] and represent a functionally immune suppressive cell population [18]. To better understand how glioma-generated MDSCs suppress T-cell populations, we evaluated T-cell proliferation in an ex vivo co-culture system. Multiple inhibitors of suspected immune suppressive mechanisms were evaluated for their ability to recover T-cell activity. Having determined that glioma-derived M-CSF and IL-34 drive M-MDSC differentiation (Figure 4) and GM-CSF and IL-6 are present in the glioma microenvironment (Figure 2A), whole bone marrow was harvested, and cells were cultured in the presence of glioma-conditioned media and exogenous GM-CSF and IL-6. The generated M-MDSCs were capable of suppressing both CD3/CD28-stimulated CD4 and -CD8 T cell proliferation (Figure 5A,B).

To identify the mechanism that glioma-induced M-MDSCs utilize to suppress T cells, a screen of inhibitors of ARG1, IFNγR2, PDL1, IDO, NOX, COX2, and A2AR were initially used to determine potential suppressive mechanisms. Arginase 1 is an enzyme that catalyzes the conversion of extracellular arginine to ornithine and urea. T cells require high levels of arginine for proliferation and functionality [33]. IFNγ signaling has been reported to mediate the immune-suppressive activity in M-MDSCs [34]. Indoleamine 2, 3-Dioxygenase catalyzes the conversion of tryptophan to kynurenine. Tryptophan metabolism is necessary for T-cell functionality [35]. NADPH oxidase (NOX) results in the generation of ROS species that can impact TCR signaling [36]. COX2 synthesizes prostaglandin E2, which inhibits T-cell activity through the upregulation of CD94 and the NKG2A complex [37]. CD73 expression, which may be expressed by M-MDSCs, converts AMP into adenosine. Adenosine has direct CD8 suppressive through interactions with the A2 adenosine receptor [31]. IL-10 induces tolerance in T cells through inhibition of the CD28 co-stimulatory pathway [38]. None of the inhibitors (tested at concentrations above their respective IC_50_) recovered the proliferation of CD4 or CD8 T cells (Appendix A); the inhibitors did not alter T cell proliferation in the absence of M-MDSCs.

Nitric oxide (NO) is a commonly reported mechanism by which MDSCs suppress T-cell activity [28]. We sought to test if inhibition of inducible nitric oxide synthase (iNOS) could recover the proliferation of T cells in the presence of MDSCs differentiated by KR158B-conditioned media. M-MDSC-enriched bone marrow-derived cells significantly suppressed the proliferation of both CD4^+^ and CD8^+^ T cells at ratios 1:2 and 1:1, respectively (Figure 5A,B). The pan NOS inhibitor (500 nM of L-NMMA) did not impact the proliferation of stimulated CD4 and CD8 T cell proliferation in the absence of M-MDSCs. In the presence of L-NMMA, CD8 T cell proliferation was recovered to 60% while M-MDSC suppression of CD4 T cell proliferation was not impacted (Figure 5A,B). These data establish that M-MDSCs utilize NOS to suppress CD8, but not CD4 T cell proliferation. This differential outcome suggests that MDSCs use different mechanisms to suppress T cell subsets.

After identifying NOS as a mechanism that M-MDSCs utilize to suppress CD8 proliferation in the ex vivo assay, we then sought to determine if iNOS is expressed by glioma-associated M-MDSCs in vivo. To this end, immunohistochemistry and flow cytometry analyses of KR158B and GL261 tumors established in *Ccr2^RFP/WT^*/*Cx3cr1^GFP/WT^* mice were conducted to assess iNOS expression in M-MDSCs. Gliomas were harvested, sectioned, and subjected to IHC analysis for iNOS expression. CCR2^RFP^/CX3CR1^GFP^ cells were found to be positive for iNOS in the tumor microenvironment (Figure 5C). Parallel flow cytometry was conducted on KR158B gliomas and spleens from tumor-bearing mice to quantify the percentage of M-MDSC that are regulating iNOS. We found that less than 20% of the M-MDSCs in the tumor microenvironment expressed iNOS while M-MDSCs in the peripheral spleen lacked iNOS expression (Figure 5D). These data imply that a proportion of M-MDSCs regulate iNOS in the glioma TME to suppress T cell activities in vivo.

### 3.6. M-MDSCs Express CD39 and CD73 in KR158B and GL261 Tumors

Since iNOS was regulated by only a fraction of M-MDSC in vivo, additional suppressive mechanisms were explored. Sarkar et.al. demonstrated that tumor-derived prostaglandin E2 (PGE2) directly induces the expression of CD73 on M-MDSCs in lung, melanoma, and breast cancer mouse models. The generation of the suppressive purine nucleoside and adenosine occurs through the degradation of adenosine triphosphate (ATP) by the combined actions of enzymes CD39 (ATP to AMP) and CD73 (AMP to adenosine) [31]. We explored this adenosine pathway in murine glioma models.

A competitive PGE2 ELISA was conducted on GL261 and KR158B-cultured cell media and tumors. Both GL261 and KR158B cells secreted PGE2. Additionally, PGE2 was found at high levels in the glioma microenvironment (Appendix A). We next evaluated the expression of CD39 and CD73 on MDSCs generated by incubating whole bone marrow cells with either KR158B cells or KR158B-conditioned media. Specifically, bone marrow was cultured in RPMI (Control), KR158B-conditioned media (75%), or co-cultured with KR158B cells and subsequently stained for MDSC markers, CD39, and CD73 after 3 days. KR158B and GL261 tumors, as well as respective peripheral tissues, were also immune profiled for MDSC markers and CD39 and CD73. Both MDSC subsets expressed CD39 on their cell surface in the glioma microenvironment and under in vitro conditions (Figure 5E,F,H, Appendix A). Percent of MDSCs expressing CD39 in bone marrow was lower than in tumor tissues or cultured conditions (Figure 5E,F,H). The more relevant marker, CD73, was expressed on less than 20% of MDSCs in the KR158B and GL261 tumor microenvironment (Figure 5E,G,I). KR158B-conditioned media and cell co-culture resulted in CD73 upregulation on the M-MDSC subset (Figure 5E,G,I). These data highlight that, similar to iNOS, the adenosine-generating pathway is regulated on a subset of M-MDSCs in the glioma microenvironment.

### 3.7. Immune Suppressive Heterogeneity Is Identified through Spatial RNAseq in Human Glioblastoma

To identify additional mechanisms of MDSC suppression, we mined the Ivy glioblastoma Atlas Project database to investigate the spatial regulation of suspected immune suppressive factors in human GBM. The Ivy Glioblastoma Atlas Project is a resource for exploring the anatomic and genetic basis of glioblastoma at the cellular and molecular level. Interest genes were queried, and Z-scores were calculated for each gene transcript level at regions of interest: (leading edge, infiltrating tumor, cellular tumor, peri necrotic zone, perivascular zone). Within the perivascular region, CCR2, CX3CR1, TGFB1, NOS3, NOS4, and ENTPD1(CD39) transcripts were enriched. NT5E (CD73), CSF ligands, CD274 (PDL1), IL1B, IL6, and IL10 transcripts were primarily found in the peri necrotic zone, whereas NOS2 (iNOS), TGFB2, and NOX1 expression is localized to cellular tumor regions. Other suspected immune suppressive factors localized to non-distinct regions within the glioma microenvironment (Figure 6A). These data support the notion that M-MDSCs utilize different pathways to exert their suppressive function and this heterogeneity could be driven by niches within the tumor microenvironment. For instance, NOS was primarily regulated in the perivascular niche while NT5E (CD73) was in hypoxic regions. This difference could help explain why the adenosine receptor antagonist (CGS-15943), targeting the CD73 pathway, was unable to recover T cell proliferation under normoxic conditions (Appendix A).

## 4. Discussion

Myeloid cells are the most abundant immune cell population in high-grade gliomas and often display an immune suppressive phenotype [39]. In this study, we found that bone marrow-derived CCR2^+^/CX3CR1^+^ cells adopt an immune suppressive cell phenotype when cultured with glioma-derived factors. Glioma-secreted CSF1R ligands M-CSF and IL-34 were determined to be key drivers of M-MDSC differentiation. Adenosine and iNOS pathways were identified and implicated in M-MDSC suppression in GBM. However, after mining a human GBM spatial RNAseq database, we hypothesized that M-MDSCs utilize a variety of different pathways to exert their suppressive function that could be driven by niches within the microenvironment. These data provide a more comprehensive, albeit complex, understanding of the mechanisms by which M-MDSCs exert the immune-suppressive function in glioblastoma.

Tumor-derived factors (i.e., IL-6, IL-10, IL-1β, PGE2, VEGF, GM-CSF, M-CSF, and IFN-γ) have been reported to induce MDSCs [32]. We utilized whole bone marrow cultured with glioma-conditioned media to show that glioma-secreted factors directly drive differentiation of monocytes into M-MDSCs. These M-MDSCs were also CCR2- and CX3CR1-expressing, implying they migrate to CCR2 ligands. Previous work from our group highlights the importance of the CCR2 axis in the migration of this functionally immune suppressive cell population [17,18]. Interestingly we saw a dose-dependent increase in CX3CR1^GFP^ expression with increasing glioma-conditioned media. One hypothesis is that M-MDSCs need to upregulate CX3CR1 to improve extravasation into the inflamed tissue. However, the implication of this result needs further investigation. A Luminex-based cytokine panel identified many differentially upregulated factors in the glioma microenvironment. This included many of the reported factors that are implicated in MDSC differentiation. To this end, we showed that CSF ligands GM-CSF, G-CSF, and M-CSF are sufficient to drive M-MDSC differentiation. We found that KR158B glioma cells secrete CSF1R ligands M-CSF and IL-34. Although we examined this phenotype in a clinically relevant, immune checkpoint-resistant model, additional mouse models should be tested for their expression of CSF ligands. Additionally, glioma cells were grown in oxygen-rich, high-glucose, and monolayer conditions. Culturing these cells in more physiological conditions might change these results significantly. We also found that CSF-1, CSF-2, and CSF-3 were all differentially upregulated in the murine glioma microenvironment. While we examined scRNAseq results from human GBM, additional studies should be conducted to discriminate glioma versus stromal contributions of these CSF ligands.

We used M-CSF neutralizing antibodies and CSF1R small molecule antagonists to block M-MDSC differentiation in KR158B-conditioned media. These results point to M-CSF and IL-34 as the primary glioma-secreted factor that drives the expansion of M-MDSCs. The TCGA database and publicly available scRNAseq datasets were used to correlate findings to human GBM. CSF-1R inhibitor PLX3397 was tested in a phase II clinical trial but did not show improvement in survival for recurrent GBM patients [40]. Quail et.al. offer an explanation for this outcome as acquired resistance to CSF-1R inhibition was correlated with increased insulin-like growth factor signaling between TAMs and tumor cells [41]. Understanding how to combat these resistance mechanisms is imperative for the use of CSF-1R inhibitors in the clinic. CSF-1R inhibition will likely have the most success in combination with existing immunotherapies. Short temporal use of CSF1R antagonists in combination with T-cell-directed therapies could unleash immune-directed anti-tumor effects while preventing resistance. It is unclear from our studies if differentiation is occurring in the bone marrow or at the site of the tumor. If differentiation is occurring within the glioma, post-monocyte extravasation, then CSF1R-targeting agents would be most effective if they cross the BBB. Approaches to improve brain penetration of small molecule compounds will be fruitful in this regard.

We also explored the mechanisms of immune suppression in CCR2/CX3CR1 M-MDSCs. Previous results from our lab established that CCR2- CX3CR1-expressing cells also display markers of M-MDSCs (Ly6G-, Ly6Chi) and are functionally immune suppressive. CCR2^+^ M-MDSCs represent a prominent infiltrating immune suppressive cell population within murine gliomas [42,43]. Their elevated presence has been shown to be correlated with a negative prognosis and poor response to prospective immunotherapy approaches such as immune checkpoint inhibitors [16,44]. Data reported here establish that CCR2^+^/CX3CR1^+^ M-MDSCs are directly involved in disrupting the proliferation and activated function of both CD4 and CD8-expressing T cells. We also found that CCR2^+^/CX3CR1^+^ M-MDSCs express iNOS in the glioma microenvironment and inhibition of NOS with L-NMMA could recover in vitro proliferation of CD8 T cells, but not CD4 T cell proliferation. This differential outcome suggests that MDSCs may use alternate mechanisms to suppress T cell subsets. However, CD4 T cells are reported to have a restricted proliferation pattern compared to CD8 T cells [45]. To this end, CD4 T cells may be more sensitive to M-MDSC suppression and require higher drug concentrations for recovery. Conversely, CD8 T cells might be more resilient at recovering from M-MDSC suppression.

Sarkar et al. showed that tumor-derived prostaglandin E2 (PGE2) directly induces the expression of CD73 on M-MDSCs and results in the production of T-cell-suppressive adenosine [31]. We explored this adenosine pathway in GL261 and KR158B models and showed a fraction of MDSC-regulated CD73. We also show in human glioblastoma that M-MDSC-suppressive pathways are heterogeneously regulated and localized to niches within the microenvironment. These results support the notion that M-MDSCs utilize numerous pathways to exert their suppressive function. Since we were incapable of blocking adenosine-mediated suppression ex vivo, our results suggest that a homogenous iNOS expressing M-MDSC is generated in vitro but a more heterogeneous population is evident in vivo. Ways to isolate enough glioma M-MDSCs for functional experimentation or better modeling the TME ex vivo may be beneficial in establishing the heterogenous immune suppressive factors that these cells utilize. This heterogeneity highlights the challenge of developing a therapeutic to target M-MDSC suppressive mechanisms. Ideally, identifying and targeting an upstream regulator could be most effective at targeting the immune suppressive mechanisms of M-MDSCs. These data provide a more comprehensive understanding of the function of CCR2^+/^CX3CR1^+^ MDSCs and further support the significance of targeting these cells in GBM (Figure 6B).

## 5. Conclusions

Our findings show that glioma-secreted M-CSF and IL-34 drive the differentiation of naïve monocytes into functionally T-cell-suppressive M-MDSCs. This suppressive phenotype is driven by the regulation of NOS. We also show in human glioblastoma that M-MDSC suppressive pathways are heterogeneously regulated and localized to niches within the microenvironment. These results support the notion that M-MDSCs utilize many pathways to exert their suppressive function in vivo. These data support the concept of the therapeutic targeting of M-MDSCs in glioma through blocking the CSF1R axis and NOS.

## Figures and Tables

**Figure 1 cancers-16-03055-f001:**
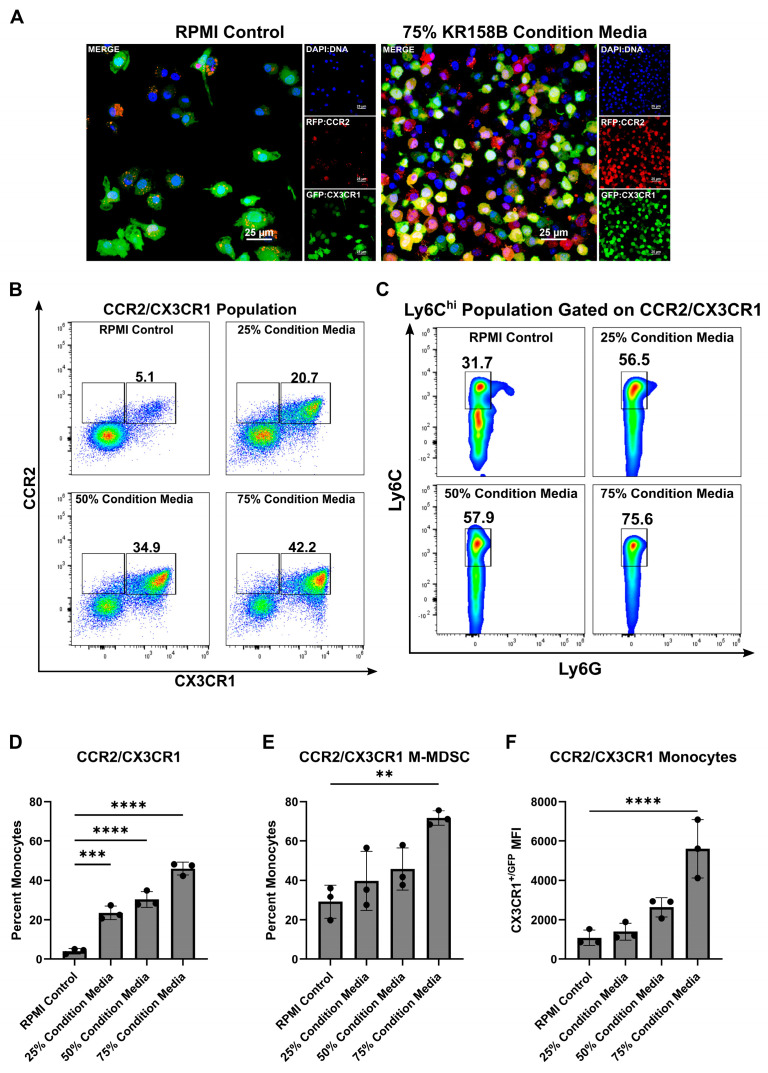
KR158B-conditioned media enriches CCR2^+^/CX3CR1^+^ M-MDSCs. Whole bone marrow from CCR2^WT/RFP^/CX3CR1^WT/GFP^ mice was cultured in the presence of KR158B glioma cell-conditioned media: (**A**) Representative fluorescent images from 1 mouse. (**B**) Representative flow cytometry plots displaying the CCR2/CX3CR1 population after gating on CD45 and CD11b (**C**) Representative flow cytometry plots displaying MDSC markers Ly6C and Ly6G after gating on CD45, CD11b and CCR2/CX3CR1. (**D**–**F**) Quantitative graphs from the previous panels. Graphs depicting the percentage of myeloid cells that are (**D**) CCR2^+^/CX3CR1^+^, (**E**) Ly6C^hi^, Ly6G^−^, CCR2^+^/CX3CR1^+^ (**F**) CX3CR1^+/GFP^ MFI of CCR2^+^/CX3CR1^+^ monocytes. (n = 3 mice). One-way ANOVA statistical analysis was conducted (Dunnett’s multiple comparisons test). Differences are compared to the stimulated control condition. *p*-values: 0.0021 (**), 0.0002 (***), <0.0001 (****).

**Figure 2 cancers-16-03055-f002:**
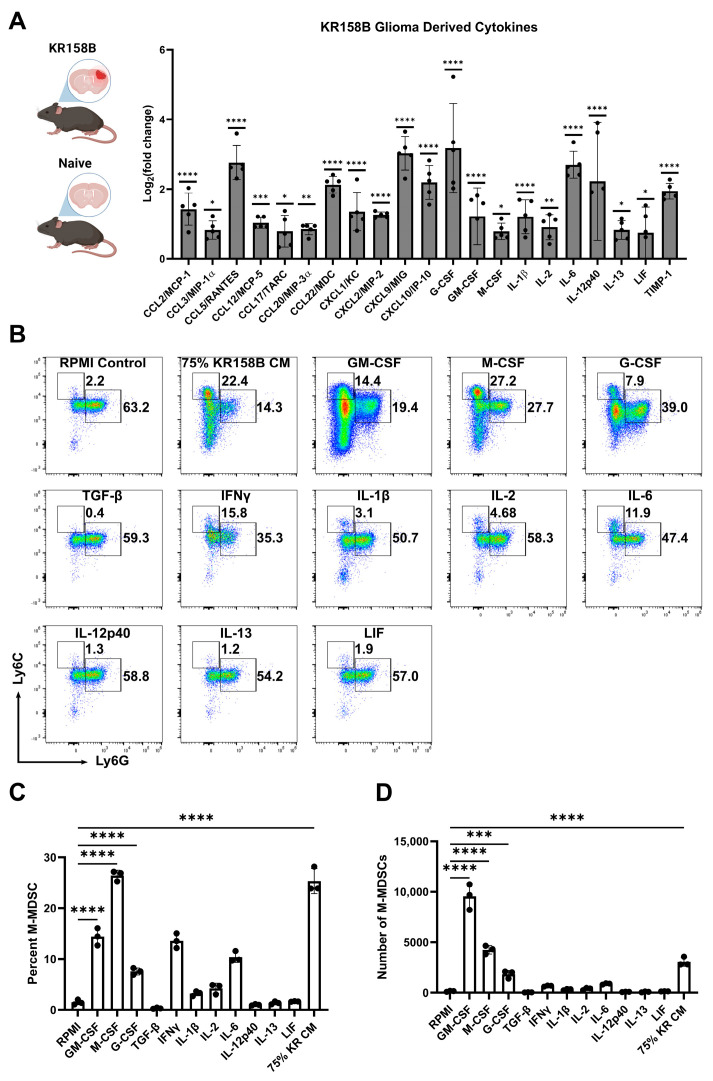
Exogenous M-CSF and GM-CSF are sufficient for M-MDSC differentiation: (**A**) KR158B gliomas (2 weeks post-implantation) and naïve brains were harvested and processed for Luminex-based cytokine analysis. Only significantly upregulated cytokines are displayed as Log2(fold change) compared to naïve samples. Data points reflect biological replicates (n = 5 mice). (**B**) Representative flow cytometry plots of Ly6C and Ly6G myeloid cells from bone marrow cultured for 3 days in 40 ng/mL cytokines or 75% KR158B-conditioned media. (**C**,**D**) Graph depicting the (**C**) percent and (**D**) numbers of M-MDSCs generated from whole bone marrow cultured for 3 days in the presence of 40 ng/mL cytokines or 75% KR158B-conditioned media (n = 3 mice). One-way ANOVA statistical analysis was conducted (Dunnett’s multiple comparisons test). Differences are compared to the control condition or between cell lines. *p*-values: 0.0332 (*), 0.0021 (**), 0.0002 (***), <0.0001 (****).

**Figure 3 cancers-16-03055-f003:**
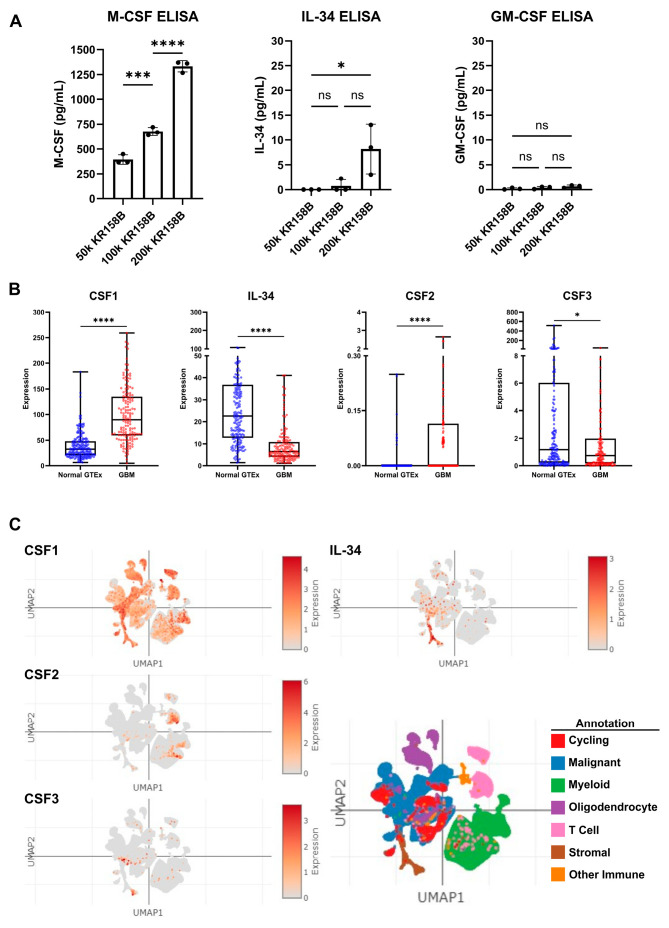
Expression of CSF ligands in murine glioma and human GBM: (**A**) Murine KR158B gliomas express the CSF ligands M-CSF and IL-34 but not GM-CSF. KR158B cells were plated at different numbers (50, 100, 200 thousand) in 24 well plates and the conditioned media were subjected to ELISA after 24 h for cytokine quantification. CSF1R ligands, M-CSF and IL-34, were found at a higher concentration in the media compared to GM-CSF (n = 3 experiments). (**B**) Human GBM expression of CSF ligands was determined from a search of the OncoDB database. TCGA dataset was queried for GBM gene expression. Results were compared to normal brain tissue from GTEx. CSF-1 (M-CSF) was differentially upregulated in the GBM microenvironment (n = 148 patients) compared to normal brain (n = 200 patients). CSF-1 was found at the highest level compared to the other CSF ligands. (**C**) Single-Cell Portal database from Broad Institute was accessed for evaluation of CSF ligand expression. The “Programs, Origins, and Niches of Immunomodulatory Myeloid Cells in Human Gliomas” dataset was used for analysis. Multiple human tumors (85) and cells (515,782) were analyzed. CSF-1 and IL34 are expressed by the malignant cell population. CSF-2 and CSF-3 (GM-CSF and G-CSF) were lowly expressed. Students’ *t*-test statistical analysis was conducted. Differences are compared to the stimulated control condition. *p*-values: >0.05 (ns), 0.0332 (*), 0.0002 (***), <0.0001 (****).

**Figure 4 cancers-16-03055-f004:**
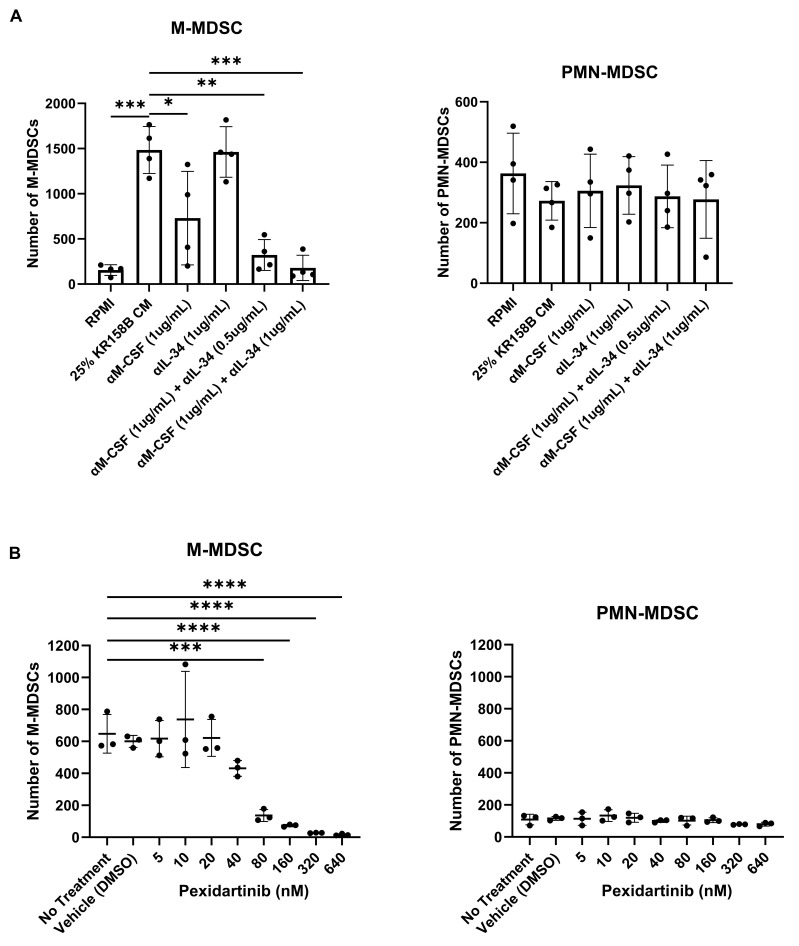
Inhibition of the CSF1R axis blocks glioma-mediated M-MDSC differentiation: (**A**) Whole bone marrow from wildtype mice (n = 3) was cultured in the presence of KR158B glioma-conditioned media with M-CSF- and IL-34-neutralizing antibodies. Highest levels of M-MDSC neutralization were observed in the conditions that combined anti-M-CSF and anti-IL-34 antibodies. PMN-MDSC numbers were not impacted. (**B**) Whole bone marrow from wildtype mice (n = 3) was cultured in the presence of KR158B glioma-conditioned media and the potent small molecule CSF1R inhibitor Pexidartinib. CSF1R inhibition resulted in a dose-dependent decrease in M-MDSCs differentiated by glioma-conditioned media. PMN-MDSC numbers were not impacted. One-way ANOVA statistical analysis was conducted (Dunnett’s multiple comparisons test). Differences are compared to the control condition or between cell lines. *p*-values: 0.0332 (*), 0.0021 (**), 0.0002 (***), <0.0001 (****).

**Figure 5 cancers-16-03055-f005:**
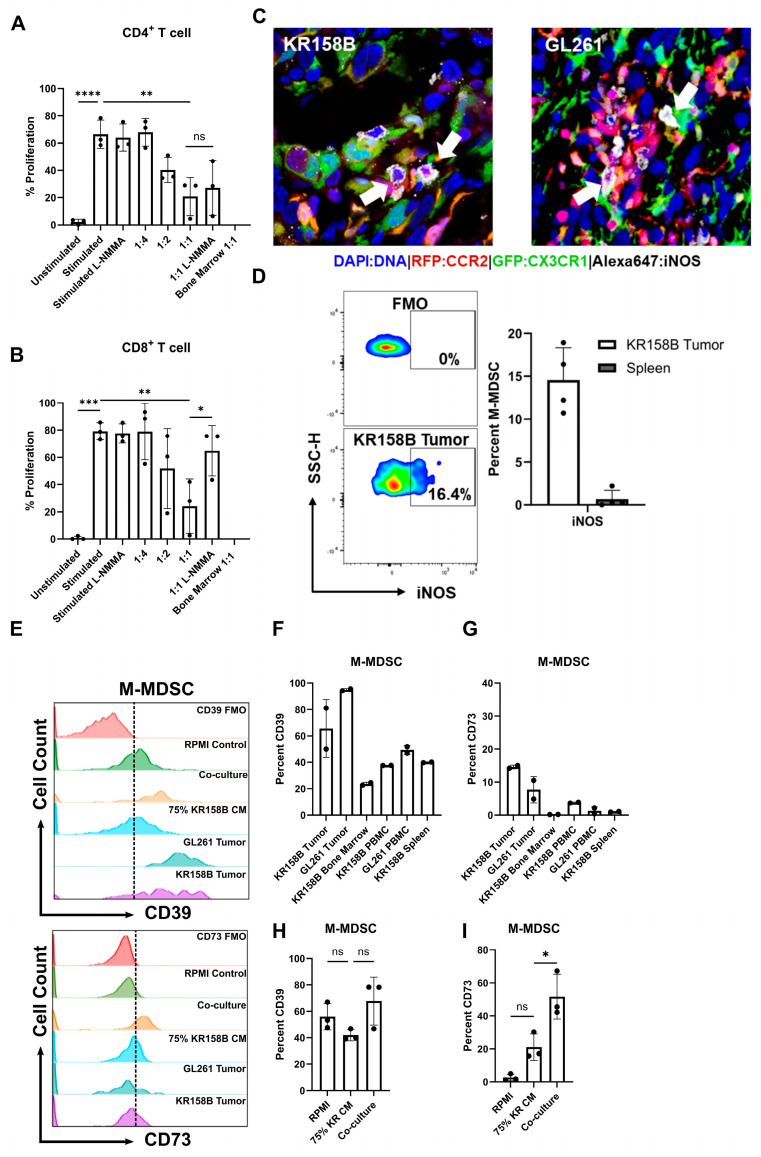
Glioma-induced M-MDSCs express iNOS, CD39, and CD73 and suppress CD8 T cells in a NOS-dependent manner. M-MDSCs were generated from whole bone marrow (n = 3) cultured with KR158B-conditioned media. T cell suppression assay was conducted in the presence of an NOS inhibitor L-NMMA (500 nM): (**A**) Graphs depicting CD4^+^ T cell proliferation being suppressed by MDSCs and cannot be recovered by L-NMMA. (**B**) Graphs depicting CD8^+^ T cell proliferation being suppressed by MDSCs and being recovered by L-NMMA. (**C**) IHC analysis of Ccr2^RFP/WT^/Cx3cr1^GFP/WT^ mice implanted with KR158B or GL261 glioma cells. Triple-positive cells (white arrows) illustrate M-MDSCs that express iNOS in the glioma microenvironment. (**D**) Representative flow cytometry plots of iNOS expression in KR158B-derived M-MDSCs. Graphs depicting percent M-MDSCs expressing iNOS in KR158B tumors or spleen (n = 4). (**E**) Representative flow cytometry histograms for CD39 and CD73 expression. (**F**,**G**) Graphs showing the percentage of M-MDSCs expressing CD39 or CD73 on cell surface in vivo (n = 2). (**H**,**I**) Graphs showing the percentage of M-MDSCs expressing CD39 or CD73 on cell surface under ex vivo conditions (n = 3). One-way ANOVA statistical analysis was conducted (Dunnett’s multiple comparisons test). Differences are compared to the stimulated control condition. *p*-values: >0.05 (ns), 0.0332 (*), 0.0021 (**), 0.0002 (***), <0.0001 (****).

**Figure 6 cancers-16-03055-f006:**
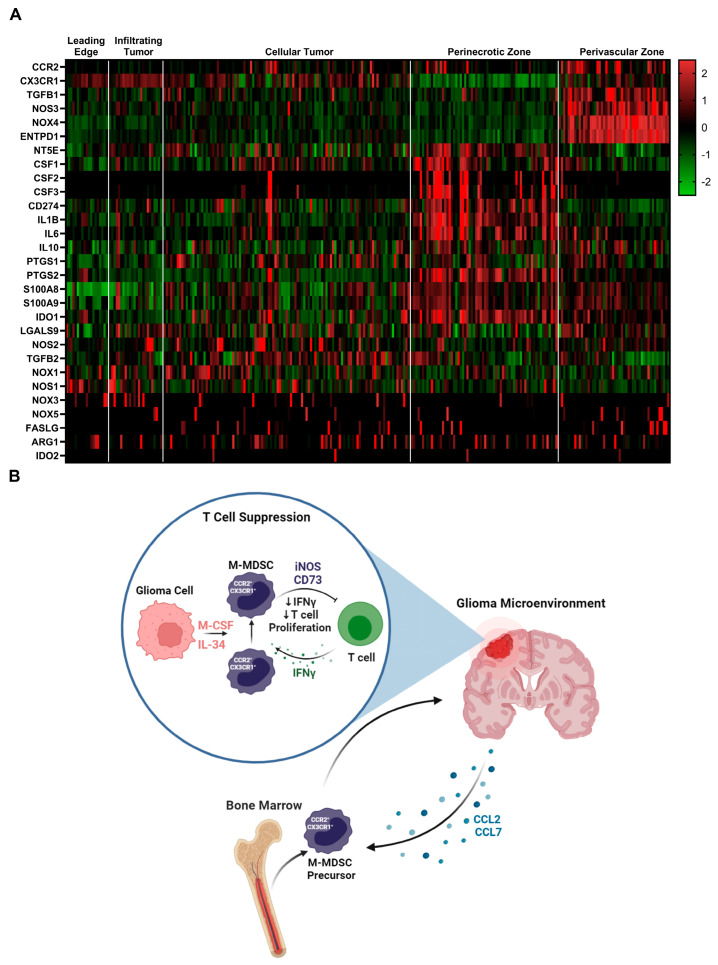
Immune suppressive heterogeneity in human glioblastoma: (**A**) Select gene expression analysis of the Ivy glioblastoma Atlas Project RNAseq database. Tumor areas queried included leading edge, infiltrating tumor, cellular tumor, perinecrotic zone, and perivascular zone. Of note, iNOS (NOS2) expression is localized to cellular tumor regions while CD73 (NT5E) transcripts were primarily found in the perinecrotic zone. CSF ligands are also present in the perinecrotic zone. (**B**) Graphical summary of CCR2^+^/CX3CR1^+^ cell mechanism in glioblastoma. Migration, licensing, and suppression mechanisms are displayed.

## Data Availability

Data supporting the findings within this study are presented within the article and are available from the corresponding author upon reasonable request.

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
