# Peer review of "CSF1R Ligands Expressed by Murine Gliomas Promote M-MDSCs to Suppress CD8+ T Cells in a NOS-Dependent Manner"

_cancers, 2024, doi:10.3390/cancers16173055_

Round 1

Reviewer 1 Report

Comments and Suggestions for Authors

The manuscript investigates the role of glioma-derived M-CSF and IL-34 in the differentiation and immunosuppressive function of monocytic myeloid-derived suppressor cells (M-MDSCs) in glioblastoma (GBM). The study identifies key mechanisms by which these factors contribute to T cell suppression and highlights the heterogeneity of the tumor microenvironment.

Major Weaknesses:

1、 The manuscript does not provide detailed information on the statistical analysis for some experiments. Including more details about the number of replicates and the statistical significance would strengthen the results.

2、 While the study identifies key drivers of M-MDSC differentiation, it does not deeply explore other potential contributing factors or their interactions.

3、 The figures are clear, but some could benefit from additional annotations to improve readability.

4、 Use "u" instead of "μ" for micro measurements, and ensure proper formatting for subscripts and superscripts throughout the manuscript.

Comments on the Quality of English Language

Use "u" instead of "μ" for micro measurements, and ensure proper formatting for subscripts and superscripts throughout the manuscript.

Reviewer 2 Report

Comments and Suggestions for Authors

Comments to authors-

The manuscript entitled “Glioma-derived M-CSF and IL-34 license M-MDSCs to suppress CD8+ T cells in a NOS-dependent manner” by Gregory P. T et al., was well written and easy to follow. I have following comments/suggestions

1.     At a glance the word “license” was not clear whether it activates or inhibits. Please use the appropriate wording to provide the best description of function

2.     Does inhibiting CCR2 using pharmacological agents effect the modulation of M-MDSC’s. Will the addition of conditioned media rescue the phenotype of pharmacological inhibition? This experiment would address the activity in a more robust manner in addition to spiking the conditioned media at the cells derived from bone marrow

3.     Perform a cluster analysis of the Luminex data based on the cytokine family to derive the patterns as the effect was brought in by various factors rather than few cytokines

4.      Please provide a segmented graph for CSF2 and CSF3 from Fig-3 panel B

5.     The ELISA graphs from Fig-3 doesn’t show statistical values, is it non-significant or the statistics were missing for this data?

6.     It would be interesting to look at the status of TH-cells in TME especially the Treg’s. Since the CD8 suppression happens, does it influence the other T-cells that could modulate the TME?

Reviewer 3 Report

Comments and Suggestions for Authors

The authors propose a role for glioma-derived M-CSF and IL-34 in activating immunosuppressive M-MDSC function. Through in vitro/ex vivo cell activation/inhibition systems and bioinformatics analysis of patient glioma data, the authors suggest that glioma-activated M-MDSCs suppress T cell (CD8) function in multiple ways including NOS pathway and CD39/CD73.

The manuscript builds on work previously published by the same and other groups and provides some new insights into mechanisms of immunosuppression in glioma. However, the impact of this data on the progress of clinical and translational glioma research is limited, as addressed in the comments below:

Major comments:

1.IL-34 levels seem to be negligible in both cell culture (Figure 3A) as well as in patient tissues (Figure 3B). So, even if it is secreted by glioma cells, no strong conclusions about its effect on activating M-MDSCs can be made. Furthermore, blocking IL-34 does not affect glioma-mediated M-MDSC differentiation (Figure 4). It is suggested that the authors revise the manuscript title, since they show conclusive data for the role of M-CSF but not for IL-34.

2. Statement in lines 424-426 and 594-595 should be revised since:

-the authors have strong data only on M-CSF

-M-CSF levels specifically do seem to affect PMN-MDSCs, where blocking M-CSF drives MDSC differentiation towards PMN instead of M forms (Supp. Figure 4B). 

3. In Fig 5B, can the authors clarify and show statistical data if the CD8 proliferation recovery in the presence of NMMA (1:1 L-NMMA) is significantly different than the suppression seen in 1:1 group?

4. From the data shown in Figure 5H and Supp. Figure 4E,F, no conclusions can be drawn about a potential role for CD73 and adenosine pathway in M-MDSCs. Additional targeted experiments (e.g using CD73 blockers/knockdown) are required to determine a functional significance.

Furthermore, 

(a)since PGE levels are minimal in glioma cells in vitro, but high in tumors grown in vivo (correlating with CD73 data), can the authors comment their thoughts on whether the in vivo tumor microenvironment influences PGE2 and ultimately CD73? Using an in vitro isolated system to study this pathway might not be the best representation

(b) MDSCs found in in vivo TME and MDSCs activated in vitro should ideally show similar phenotypes, can the authors hypothesize why MDSC activation in vitro in the presence of glioma conditioned media is unable to activate CD73 expression? It is suggested the authors test M-MDSC activation in direct contact with glioma cells (KR158B and GL261) along with testing glioma conditioned media. 

5. From the in vivo experiments conducted, can the authors clarify the percentage of licensed M-MDSCs in the TME of KR158B/GL261 glioma bearing mice, compared to the bone marrow/circulation?

6. The data would be further strengthened by co-staining T cell populations (active/inactive CD4, CD8, Tregs) and M-MDSCs within the in vivo glioma tumor tissues to show immunosuppression in the presence of activated M-MDSCs.

Comments on the Quality of English Language

The quality of English language is sound throughout most of the manuscript with minor corrections required:

1. Correct the spelling to "queried" in line 277

2. Correct the spelling to "dose" in line 313

Round 2

Reviewer 2 Report

Comments and Suggestions for Authors

Thank you for addressing most of the comments, I have the following comments.

1.         At a glance the word “license” was not clear whether it activates or inhibits. Please use the appropriate wording to provide the best description of function

We thank the reviewer for this comment and agree that the term ‘license’ could be unclear. However, this nomenclature has precedent set by Condamine and Gabrilovich (https://www.ncbi.nlm.nih.gov/pmc/articles/PMC3053028/). Below is an example model that describes the licensing phase followed by activation by inflammatory cytokines. Since CSF1R ligands are not regarded as inflammatory cytokines we believe we should not use the word “activate” or “promote” as our results only suggest that we have at least a licensed cell that will become suppressive or already is. We felt that the word license was a conservative word that aligns with the literature.

The explanation doesn’t still address the use of license word in the main heading. The T Condamine etal manuscript only talks about function and it never used the term license in their entire manuscript. Activate or promote relate to the function of a receptor rather than the ligand, function of a ligand can always lean towards both sides based on the receptor it binds. So, there is no harm in identifying the function as a promote or activate or suppress based on the downstream response from the receptor rather than ligand itself. Best example is GPCR, serotonin, choline and dopamine can act as both activating and inhibiting based on receptor type (Gs, Gi, Gq) it binds, the response is labelled on the basis of receptor but not on the basis of ligand. I would encourage you to modify it accordingly otherwise it would be confusing to the readers

3. Perform a cluster analysis of the Luminex data based on the cytokine family to derive the patterns as the effect was brought in by various factors rather than few cytokines

We thank the reviewer for this comment. However, it is unclear what type of cluster analysis should be conducted on this dataset. Additional information in this regard would be helpful to conduct the proper analysis if needed. As presented, it shows all of the differentially upregulated cytokines in the glioma microenvironment compared to naïve brain tissue. Chemokines, colony stimulating factors, and interleukins are currently grouped together.

I’m sorry if I was not clear previously. Grouping the cytokines and chemokines based on the receptor function/family and performing an HCA would help to understand the patterns and derive better conclusions.

Author Response

  1. The explanation doesn’t still address the use of license word in the main heading. The T Condamine etal manuscript only talks about function and it never used the term license in their entire manuscript. Activate or promote relate to the function of a receptor rather than the ligand, function of a ligand can always lean towards both sides based on the receptor it binds. So, there is no harm in identifying the function as a promote or activate or suppress based on the downstream response from the receptor rather than ligand itself. Best example is GPCR, serotonin, choline and dopamine can act as both activating and inhibiting based on receptor type (Gs, Gi, Gq) it binds, the response is labelled on the basis of receptor but not on the basis of ligand. I would encourage you to modify it accordingly otherwise it would be confusing to the readers

We thank the reviewer for this very detailed feedback and agree with the nomenclature change to “promote”. The manuscript now reflects this change and the word license has been removed.

  1. I’m sorry if I was not clear previously. Grouping the cytokines and chemokines based on the receptor function/family and performing an HCA would help to understand the patterns and derive better conclusions.

The Luminex analysis was limited to a small number of genes/proteins and only identified a few chemokines/cytokines that were upregulated in tumor tissue compared to naïve brain. The graph displaying these results (Fig 2A) does indeed cluster them by families. Eleven of them are chemokines and the chemokines are sorted by subfamily (7 CC and 4 CXC). CSF1R ligands are also grouped together which leaves a small number of miscellaneous cytokines. The primary outcome of this data allowed us to formulate a hypothesis regarding which specific cytokines, present within the tumor microenvironment, were important for driving the presence and/or immune-suppressive phenotype of monocytic myeloid-derived suppressor cells. As such, we performed the experiment described in panels B-D of Figure 2, where we examined the effect of many of these individual cytokines on myeloid cell sub-populations generated from murine bone marrow. Given the limited number of cytokines and chemokines analyzed, with many of them having unknown and potentially pleiotropic functions within the glioma microenvironment, it is unclear to us how a hierarchical cluster analysis will add value to the dataset and the conclusions drawn from these results.